



# Possible controls on Arctic clouds by natural aerosols from long-range transport of biogenic emissions and ozone depletion events

Rupert Holzinger[1], Oliver Eppers[2,3], Kouji Adachi[4], Heiko Bozem[2], Markus Hartmann[5,a], Andreas Herber[6], Makoto Koike[7], Dylan B. Millet[8], Nobuhiro Moteki[7], Sho Ohata[7,b,c], Frank Stratmann[5], Atsushi Yoshida[7,d]

[1]Institute for Marine and Atmospheric Research, Utrecht (IMAU), Utrecht University, The Netherlands
[2]Johannes Gutenberg University of Mainz, Institute for Atmospheric Physics, Mainz, Germany
[3]Particle Chemistry Department, Max Planck Institute for Chemistry, Mainz, Germany
[4]Department of Atmosphere, Ocean, and Earth System Modeling Research, Meteorological Research Institute, Tsukuba, Japan
[5]Atmospheric Chemistry Department (ACD), Leibniz-Institute for Tropospheric Research (TROPOS), Leipzig, Germany
[6]Alfred Wegener Institute Helmholtz Centre for Polar and Marine Research (AWI), Bremerhaven, Germany
[7]Department of Earth and Planetary Science, Graduate School of Science, The University of Tokyo, Tokyo, Japan
[8]University of Minnesota, Saint Paul, MN, USA
[a]now at: Atmospheric Science, Department of Chemistry and Molecular Biology, University of Gothenburg, Gothenburg, Sweden
[b]now at: Institute for Space–Earth Environmental Research, Nagoya University, Nagoya, Aichi, Japan
[c]now at: Institute for Advanced Research, Nagoya University, Nagoya, Aichi, Japan
[d]now at: National Institute of Polar Research, Tokyo, Japan

*Correspondence to*: Rupert Holzinger (r.holzinger@uu.nl)

**Abstract.** During the PAMARCMiP 2018 campaign (March and April 2018) a proton-transfer-reaction mass spectrometer (PTR-MS) was deployed onboard the POLAR 5 research aircraft and sampled the high Arctic atmosphere under Arctic haze conditions. More than 100 compounds exhibited levels above 1 pmol/mol in at least 25% of the measurements. We used back trajectories and acetone mixing ratios to identify periods with and without long-range transport from continental sources.

Air masses with continental influence contained elevated levels of compounds associated with aged biogenic emissions and anthropogenic pollution (e.g., methanol, peroxyacetylnitrate (PAN), acetone, acetic acid, methylethylketone (MEK), proprionic acid, and pentanone). Almost half of all positively detected compounds (>100) in the High Arctic atmosphere can be associated with terpene oxidation products. This may constitute a signature of biogenic terpenes and their oxidation products on the high Arctic atmosphere. Many of these compounds will condense and produce biogenic secondary organic aerosol (SOA) – a natural source of organic aerosol (OA) in addition to the aerosols that can be associated with pollution.





Therefore, we hypothesize that biogenic SOA may have exerted significant control over the complex system of aerosols, clouds and longwave radiation in the pre-industrial Arctic winter, even though their role is likely marginal under
contemporary polluted Arctic haze conditions. However, biogenic SOA may become an important factor in the futRure again, if biogenic emissions are enhanced due to climate change and if polluting technologies are phased out in the future. During two flights, surface ozone depletion events (ODE) were observed that coincided with enhanced levels of acetone, and methylethylketone. There is evidence that ODEs may also be associated with the emission of biogenic ice-nucleating particles (INP) because the filter samples taken during these two flights exhibited enhanced levels of highly active INP.
Both these processes, INP production in association with ozone depletion events, and the transport of biogenic SOA could require corrections in estimates of Arctic change. If the preindustrial effects from these natural factors was stronger than is thought, subsequent climate changes over the Arctic may be larger than currently assumed.

## 1 Introduction

The Arctic region is warming more strongly and faster than the rest of the Eart's surface due to complex feedback mechanisms that cause the so-called Arctic amplification. These feedback loops can be driven by the warming itself or by different types of pollution. Pollution accumulates during the winter in the Arctic region and produces Arctic haze which has the potential to alter cloud properties and thus significantly alter precipitation and the longwave heat balance during the Arctic night. With sunrise in spring the shortwave component adds further complexity. There are no measurements of
aerosol and cloud properties during the preindustrial Arctic night making the direct quantification of change an impossible task. However, it is possible to infer the change with good understanding of the undisturbed system. Below we report evidence for natural processes that may have strongly controlled High Arctic clouds, precipitation and longwave radiation balance during the Arctic winter and spring before anthropogenic pollution became the dominant control.
In this study we evaluate measurements of atmospheric trace gases in the high Arctic atmosphere during two weeks in
March/April 2018 performed onboard the POLAR 5 research aircraft during the PAMARCMiP (Polar Airborne Measurements and Arctic Regional Climate Model simulation Project) campaign (Wesche et al. 2016, Herber et al. 2012). The campaign was conducted after the polar sunrise but still under Arctic haze conditions. This can be seen, for example, in Pernov et al. (2021), who analysed gas phase organics measured at Villum Research Station (VRS)/Station Nord (81.6°N, 16.7°W) in the period April through October, 2018, and report an Arctic haze factor that dominated until the end of April but
was marginal for the rest of their campaign. The difference between the first few weeks and the rest of their study period is also evident when considering the measured mixing ratios of benzene, an anthropogenic pollutant from fossil fuel and biomass burning. In April Pernov et al. (2021) reported benzene values around 80 pmol/mol, whereas levels stayed below 20 pmol/mol during the summer (June through August). This is strong evidence for two fundamentally different regimes that govern the chemical composition of the High Arctic atmosphere: (i) long-range transport of pollution and absent local





photochemistry during winter Arctic haze, and (ii) stronger influence from local marine/sea-ice sources and active photochemistry when the polar dome is more isolated and thus long-range import of pollution is reduced. These two regimes also manifest in the regional chemical composition of organic aerosol (OA). Nielsen et al. (2019) report measurements with a soot particle aerosol mass spectrometer covering 93 days in the period Feb-May, 2015, at VRS. They found that OA contributed 24% to the total aerosol burden. A positive matrix factorization (PMF) analysis revealed a transition between a

component tagged 'Arctic Haze Organic Aerosol' that was active until mid/end April and a 'Marine Organic Aerosol' component that started emerging early in April and dominated OA in May.

Considering the 'Arctic haze' factors reported by Pernov et al. (2021), and the timing of the PAMARCMiP campaign, we conclude that the Arctic haze regime dominated during the campaign, but early signs of the transition to the summer regime were likely present as well.

**2 Materials and Methods**

The PAMARCMiP campaign was conducted from Villum Research Station /Station Nord (81.6°N, 16.7°W), Greenland, during March and April 2018. The POLAR 5 research aircraft sampled the Arctic atmosphere during 14 flights that covered a region between 78°N -85°N, 20°W -20°E between 50-5000 m above sea level (Herber et al., 2019).

The PTR-MS (PTR-TOF1000, with hexapole ion optics, Ionicon Analytik GmbH, Austria) was mounted abord the POLAR

5 aircraft and sampled from a shared inlet that consisted of backward facing PFA tubing ( length 3 m, ID 9.55 mm ) that was flushed with a constant flow of ~20 L/min. The main airstream was directed to the $CO_2/H_2O$, CO, and ozone instruments with a ~70 ml/min subflow pulled to the PTR-MS through two meters of PEEK tubing (ID 1 mm). The PTR-MS drift tube (and inlet system) was heated to 60°C and operated at a pressure of ~2.8 hPa, and a reduced electric field strength (E/N) of 130 Td. The ion source was supplied with a (gaseous) water flow of 5.5 mL/min (standard conditions, 1013 hPa, 0°C) that

was maintained by a flow controller pulling from the headspace of a reservoir with liquid water. At room temperature, the water saturation vapor pressure is sufficient to maintain the flow, but with cooler temperatures less water vapor is provided to the PTR-MS ion source.

Between flights, all instruments were powerless and the cabin temperature of the POLAR 5 aircraft could be kept above freezing with all available external heating. So, the high vacuum in the PTR-MS could not be maintained between flights,

which resulted in relatively high and variable background contamination. Approximately one hour before take-off the instruments were powered up and cabin temperatures reached ~20°C during the flight, however, for most flights the water flow into the PTR-MS ion source was not stable during the first hour after take-off. Another issue was encountered at flight levels above ~4200 m when the ambient air pressure became too low to provide sufficient flow into the instrument, which resulted in lower drift tube pressure and higher E/N values. All deviations from standard operation conditions of the PTR-

MS reduced the accuracy and quality of the measurements and therefore we focus here on the periods when the instrument performance was at its best –rejecting the first part of all flights and all data collected more than ~4000 m above sea level.





Figure 1 shows a one-hour segment of acetone measured during flight 2 (March 25, 2018) to illustrate the operation and performance of the instrument. Ambient air was monitored continuously for 180 s while recording data at a frequency of 1 Hz. After every ambient air period a zeroing cycle was performed to determine instrumental noise and background contamination. Zeroing was done for 20 s (but every 4[th] time for 40 s, see Figure 1) by directing the flow through a catalyst that consisted of a 14 cm long stainless steel tube (ID 9 mm), filled with platinum coated quartz wool, and heated to 350°C. Switching to the catalyst was done with a low internal volume multiport valve (Valco Inc., stainless steel with sulfinert© coating, heated to 60°C). Volume mixing ratios have been calculated using the PTRwid software (Holzinger 2015) and the kinetic approach described, for example, by Hansel et al. (1995). The transmission was calibrated by Ionicon Analytik GmbH in December 2017 (together with the installation of the hexapole upgrade) and a gas standard (Apel and Riemer for details see Holzinger et al., 2019) was used to confirm the stability throughout the campaign. For compounds contained in the gas standard we used the reaction rate constants specified in Holzinger et al. (2019), for all other compounds we used a default reaction rate constant of 3 x $10^{-9}$ cm$^3$ s$^{-1}$ molecule$^{-1}$. The instrument featured a sensitivity in the range 400-600 counts per second for mixing ratios at the 1 nmol/mol level, and a mass resolution capacity of 1200 (FWHM). PTRwid processed a unified masslist (Holzinger 2015) containing 353 ions and for all these ions the signal was extracted and volume mixing ratios were calculated at a 1 second time resolution.

After basic data processing, we produced 10 second averages. This step significantly improved the precision of the data. E.g. we expect a signal of ~20 counts per second from a compound present at 40 pmol/mol in the atmosphere. The statistical uncertainty of the data at 1 s time resolution is thus 22% (i.e. sqrt(20)/20) and is reduced to 7% (i.e. sqrt(200)/200) when considering a 10 s time resolution. So, from 180 s of ambient air sampling we formed 18 values and from the zeroing periods we formed 2 or 4 values at 10 s time resolution. In order to subtract instrumental noise and background contamination we calculated a linear fit through the neighbouring 4 or 6 zero measurements and pointwise subtracted the fitted value from the ambient air signal. The linear fit was also used to de-trend the zeroing values that were separated by the 3 minutes of ambient air sampling. The limit of detection (LOD) was calculated as three times the standard deviation of de-trended 4 or 6 zero measurements. The final dataset contains 7771 values for 353 ions at 10 s time resolution, roughly corresponding to 25 hours of measurement.

Supplement table S1 lists campaign mean volume mixing ratios together with the 25%, 50%, and 75% quantiles for all ions that exhibited signals above the LOD for more than 3% of the 7771 measurements. Supplement Figure S1 shows the signal detected at 143.109 Th during all flights as an example to illustrate how the 3% threshold has been calculated.

**Auxiliary measurements**

A detailed description of the CO and CO$_2$/H$_2$O instrument can be found in Bozem et al (2019). For CO measurements, regular calibrations with a NIST traceable CO standard and zero measurements were used to determine (and correct) for instrument drifts. For PAMARCMiP, the precision of the calculated CO mixing ratio is 1.72 nmol/mol. Similar to CO, calibrations for the CO$_2$/H$_2$O instrument were performed in time intervals of 15 to 30 min using a NIST traceable calibration





gas with a known $CO_2$ mixing ratio and $H_2O$ close to zero. For the campaign, the precision of the instrument is given as 0.04

ųmol/mol for $CO_2$ and 1.7 ųmol/mol for $H_2O$. The temporal stability was calculated from the mean instrumental drift and

was estimated with 0.42 pp ųmol/mol mv for $CO_2$ and 21.6 ųmol/mol for $H_2O$. Thus, the total uncertainty for $CO_2$ and $H_2O$

amounts to 0.43 ųmol/mol and 22.0 ųmol/mol, respectively.

Ozone ($O3$) mixing ratios were measured using UV absorption at 254 nm with a Dual Beam Ozone Monitor 205 (2B

Technologies). UV light passes two separate 15 cm long absorption cells, which are flushed alternately with ozone-filtered

and ozone-unfiltered air. The ozone mixing ratios were derived by measuring the absorption ratio as the ratio of the

respective intensities for the unfiltered and filtered case. The total uncertainty of the ozone mixing ratios is determined by the

instrumental precision and amounts to 1.21 nmol/mol.

Black carbon (BC) and non-absorbing particles (non-BC) in the size range 75-850 nm and 185-850 nm, respectively, were

measured by a single-particle soot photometer (SP2; Droplet Measurement Technologies (DMT), Longmont, CO, USA)

operated by The University of Tokyo (Yoshida et al., 2020). More information about these measurements can be found in

Ohata et al. (2021).

**Back trajectories**

The model LAGRANTO (Wernli and Davies, 1997; Sprenger and Wernli, 2015) was used to calculate backward trajectories

for the sampled air masses. As meteorological input, operational data from the European Centre for Medium-range Weather

Forecast (ECMWF) with 0.125° horizontal resolution and 137 vertical hybrid sigma-pressure levels were used. Trajectories

were initialized every 10 s from coordinates along individual research flights and calculated 10 days backward in time. The

resolution of the individual output trajectory is 1 h.

**3 Results and Discussion**


**No extensive plumes**

Figure 2 shows flight tracks, back trajectories, and the mixing ratios of acetone for the 10 flights during which the PTR-MS

collected high quality data. The trajectories show that with a few exceptions the sampled air remained within the Arctic cycle

for at least 5 days prior to sampling. Acetone mixing ratios remained in the range 300-1000 pmol/mol and clear variations

could be observed. The low acetone mixing ratios suggest that no pollution plumes from biomass burning or other

anthropogenic sources have been detected. This idea is supported by stable mixing ratios around 100 pmol/mol of benzene

(Figure 3), which is a tracer for all combustion related sources. The observed levels of benzene are typical for Arctic haze

conditions and similar levels have been observed at VRS in spring 2018 (Pernov et al., 2021). Figure 3 reveals that

occasionally local plumes from VRS and/or the adjunct military camp Station Nord have been captured towards the end of

several flights (see time stamps 31.74, 2.59, and 3.7 in Figure 3). Toluene is another tracer of combustion sources with





similar emission factors as benzene (e.g. Christian et al. 2003, Holzinger et al. 2001). However, the detected mixing ratios of toluene were only ~10 pmol/mol (Supplement Figure S2), which reflects the five times faster reaction with OH than benzene. The ratio benzene/toluene is typically above 5 which shows that the sampled air was photochemically well aged – aside from the local plumes referenced above. The absence of non-local pollution plumes is in agreement with the low black carbon concentrations that were reported by Ohata et al. (2021) for the same campaign. The same authors also report biomass burning plumes; however, these plumes were detected at altitudes above 4500 m and are thus outside the periods evaluated here. Adachi et al. (2021) analysed aerosol filter samples by transmission electron microscopy, and, likewise, they found biomass burning signatures only on filters taken above an altitude of 4000 m.

Acetone mixing ratios exhibit a distinct variation (Figure 2). The leading assumption is that higher acetone mixing ratios are observed in aged continental air masses that contain biogenic emissions and/or anthropogenic pollution. We will test this hypothesis in the following section.

**Identification of air masses with and without recent continental influence**

Benzene mixing ratios exhibited the largest variability during the flight of March 25th, 2018 (Flight 2, top left chart in Figure 3). The trajectory analysis also reveals an air source region covering northern Europe to east Asia (Figure 2). The top panels of Figure 4 show seven day back trajectories and mixing ratios of acetone, CO, $CO_2$, ozone, and water vapor during flight 2. Three periods for which trajectories are plotted are specified on the top right panel in green, blue, and magenta, respectively. During the period marked in green, the highest acetone mixing ratios were measured. The trajectory analysis of this period shows that the air was lifted seven days before sampling above Scandinavia and moved at ~3000 m altitude across Siberia to the location of sampling. The period marked in blue also exhibits high acetone mixing ratios. The trajectories move beyond 50°N at high altitudes. Most trajectories have ground contact somewhere between Siberia and the Pacific Ocean to the east of Japan. The period marked and plotted in magenta is a short episode during the 'blue' period where air with much lower acetone mixing ratios (400 pmol/mol) was sampled. These seven-day back trajectories (plotted in magenta in the top left panel) have no ground contact and they do not extend beyond 65°N. These case studies show that higher acetone concentrations can be associated with continental influence.

The bottom panels of Figure 4 present a case study for flight 7, which was a transfer flight from Longyearbyen, Svalbard, to Station Nord, Greenland, on March 28, 2018. During the first part of the flight (marked in green), the acetone mixing ratios were low, around 300 pmol/mol, and then steeply rose to a high plateau around 900 pmol/mol. The trajectory analysis revealed that the low acetone mixing ratios were associated with air that moved close to the surface from Svalbard westwards across the North Atlantic Ocean before encountering the Greenland ice sheet. The ice sheet forced the air upwards and northwards to an altitude of ~2500-3000 m above sea level, followed by an eastward deflection at the same altitude towards the sampling location. Extending the analysis to 10 days did not reveal any signs of continental influence.



Back-trajectories for the elevated acetone period (marked in blue in the right hand bottom panel of Figure 4) show that these air masses originate from North Canada, but the trajectories indicate no surface contact within the past 5 days. In this case,

however, the measured VOCs clearly reveal continental influence on the sampled air, despite missing detection of surface contact in the trajectory analysis. For example, enhanced levels of methanol, PAN, MEK, and pentanone (Figures S3, S4, S11, and S14, respectively) clearly show the continental signature, while low levels of acetonitrile rule out biomass burning as a source of enhanced VOC concentrations. The usefulness of acetone as tracer for continental influence will be further corroborated below, but before that we show cases that demonstrate other processes in the Arctic ecosystem.

**Sea ice emissions during ozone depletion events**

The case studies depicted in Figure 5 show periods of high acetone mixing ratios that cannot be related to continental influence but are linked with emissions from sea ice. The panels of Figure 5 show trajectories, flight altitude, and mixing ratios of acetone, CO, $CO_2$, ozone, and water during flight 8 and 9 on March 30th and 31st, 2018, respectively. All trajectories shown do not suggest continental influence. Even when extended to 10 days, the air mass origin remains largely

north of 65°N and above the sea. Flights 8 and 9 were dedicated to measurements of sea ice thickness and included very low flight legs at 50 m above the surface that intercepted with slightly higher legs at 120 m above the surface. The low ozone concentrations (see right hand panels in Figure 5) reveal that an ozone depletion event (ODE) was active during these two flights. These events, first reported by Oltmans (1981), have been associated with bromine emission from freshly formed sea ice (see Simpson et al., 2007 and references therein). During flight 8, an inverse relation between ozone and acetone is

obvious. The ODE was very shallow during the first part of flight 9 (before 31.55), where acetone was observed to decrease by ~200 pmol/mol whenever the aircraft climbed from 50 m to 120 m. Both features clearly confirm the emission/production of acetone during ODEs. Yokouchi et al. (1994) reported an inverse correlation between ozone and acetone during ODEs, and Guimbaud et al. (2002) suggested acetone production in the snow pack because the gas phase photooxidation of propane is insufficient to explain the observed levels. Halogen chemistry in the Arctic troposphere is an active field of research

(Simpson et al. 2015, Summer et al. 2002, Grammas et al. 2002, Dassau 2004) and fast oxidation pathways have been established with products such as formaldehyde, acetaldehyde, acetone, methylethylketone (MEK), and peroxyacetylnitrate (PAN), but the balance between production and loss is often fragile. For example, Hornbrook et al. (2016) observed enhancement in acetone, MEK, and pentanone, while aldehydes such as formaldehyde and acetaldehyde were depleted during ODEs. Our measurements confirm the emission/production of MEK and pentanone (see Figures S11 and S14,

respectively) during ODEs, exhibiting enhancements that were approximately 35% and 3% of the acetone enhancement, respectively, on a molar basis. For formaldehyde, acetaldehyde and PAN we were not able to detect production or destruction during ODEs, which was due to technical reasons for the aldehydes (poor sensitivity due to high background contamination around the mass to charge ratios of 31 and 45 Th, respectively). Dassau et al. (2004) analysed PAN measurements during ODEs in Alert, Canada, and suggest that PAN production due to fast radical chemistry in the gas phase





is balanced by snow pack deposition. So, the PAN mixing ratios may have been stable due to the balance between loss and production during the ODE observed here.

It is worthwhile pointing out that Hartman et al. (2020) detected enhanced concentrations of ice nucleating particles (INP) that are ice-active at high temperatures during three flights of the PAMARCMIP campaign. With onset temperatures as high as 265.65 K, these enhancements were most pronounced during the two flights with the observed ODE events discussed

here. The third flight with INP enhancement (onset temperature below 263 K) was flight 2 (discussed above) where back trajectories suggest continental influence from Scandinavia, Siberia, and east Asia. Si et al. (2019) report INP concentrations in the Canadian High Arctic during spring at onset temperatures of 248.15, 253.15, and 258.15 K, respectively. They found that only the 248.15 K INP were correlated with mineral dust – which is known for its capacity to facilitate ice nucleation. Therefore we assume that other types of INP were collected on the filters during the PAMARCMiP campaign, and

consequently Hartmann et al. (2020) hypothesize about a local biological process as a source for the observed INP. Huang et al. (2021) provide an overview of biological ice nucleating particles in the atmosphere. They highlight the ice nucleating capacity y of biological macromolecules (>20 kDa). These structures typically possess areas with very high ice nucleation activity (INA) in the way that hydrophobic and hydrophilic functional groups alternate at distances typical for ice lattice. Condensing water molecules are thus ordered in a way that promotes freezing. Through such a mechanism, biological

material can promote ice nucleation at much higher temperatures than other solid surfaces such as mineral dust. A biogenic contribution to the bromine emission during ODE events has been suggested for a long time (Barrie et al. 1988, Sturges et al., 1992), so in that respect the co-emission of biological INPs seems plausible.

**Transport of biogenic compounds to the high Arctic atmosphere**

Table 1 shows mean mixing ratios of a number of organic compounds that exhibited mixing ratios above the limit of detection during the periods marked in Figure 4 and 5. As discussed above, these air masses were classified as continentally influenced (time stamps: 25.67 - 25.68, 25.70 - 25.715, 28.67 - 28.685), Arctic/marine (time stamps: 28.64 - 28.65, 30.34 - 30.355, 31.64 - 31.675, 31.71 - 31.72), and sea ice/ODE (time stamps: 30.375 - 30.46 except 30.4 - 30.42, 30.4 - 30.42, 31.53 - 31.625). Figures 2, 3, and S2-S14 show measured mixing ratios during all flights for the compounds shown in Table

255    1.

The levels of benzene and toluene are statistically not different between the three classifications (95% confidence level, 1-tailed student t-test, homoscedastic distribution), reflecting similar levels of (aged) anthropogenic pollution in all air masses. Formic acid, DMS, and $C_4H_9^+$ are also compounds that do not exhibit significant difference. The latter ($C_4H_9^+$) can be associated with anthropogenic hydrocarbon pollution, while high background concentrations and the resulting noise is the

likely reason that possible differences could not be resolved for formic acid and DMS. Note that the average mixing ratio of formic acid is still two times higher in continentally influenced air, which is in agreement with Stavrakou et al. (2012) who





report satellite evidence for a large source of formic acid from boreal forests. All other compounds listed in Table 1 exhibit significantly higher concentrations in air masses with continental influence compared to air classified as 'Arctic/marine'.

Methanol, acetone, acetic acid, isoprene, MEK, propionic acid, and pentanone can be associated with primary and secondary biogenic emissions (e.g. Oderbolz et al. 2013, Lee et al. 2021), but carbonyls are also produced from photochemical degradation of anthropogenic hydrocarbon pollution. PAN is a stable reservoir of reactive nitrogen that is produced from primary $NO_x$ emissions. $C_2H_3O^+$ is a known fragmentation product in the PTR-MS that is produced from acetic acid with a yield of ~30%. However, the observed levels are about twice as high as the observed signal at 61.03 Th (acetic acid), which suggests that most of this signal must be attributed to other compounds. Possible candidates include thioacetic acid, pyruvic acid, and alkyl acetates (Pagonis et al. 2019), but there are likely additional compounds that produce this ion. $C_6H_{11}^+$ can be tentatively attributed to cis-3-hexen-1-ol, which is a leaf alcohol that has been associated with leaf wounding (Fall et al., 1999).

In Figure 6a-p we show average and median concentrations of anthropogenic (a-h) and biogenic (i-p) components categorized in bins of increasing acetone mixing ratios. Note that for this analysis we removed periods during which ozone was below 30 nmol/mol to avoid any bias from ODE events. All components exhibit the highest mixing ratios in either of the two bins with the highest acetone concentrations. For most components both, average and median, concentrations in these two bins are higher than in the three bins with acetone levels below 600 pmol/mol. This analysis further supports the idea that acetone serves as tracer for continental influence. Anthropogenic pollutants such as black carbon, ozone, PAN are clearly enhanced under conditions of high acetone levels. The enhancement is less clear for relative stable pollutants such as CO and benzene, likely due their accumulation during winter Arctic haze. Predominately biogenic compounds such as methanol, formic acid, acetic acid, and the $C_6H_{11}^+$ fragment from cis-3-hexen-1-ol, are clearly enhanced as well. Isoprene and its oxidation product MVK+MACR also exhibit an increase with higher acetone levels, however, less pronounced which is likely due to their shorter lifetime. It is remarkable that the signal measured at 137.1 Th (attributed to monoterpenes) and 139.1 Th (attributed to nopinone - a ozonolysis product of β-pinene) are enhanced as well. Considering an ozone concentration of 30 nmol/mol, transport at 253 K and 800 hPa, β-pinene, 3-carene, and α-pinene are the most stable monoterpenes with a lifetimes of ~1.6, 0.6, and 0.3 days, respectively (Khamaganov and Hites, 2001). Assuming that the 0.2 pmol/mol increase of median monoterpene mixing ratios (shown in Figure 6-o) is due to β-pinene, and considering a transport time of 7 days, the mixing ratio at the region of origin should be ~16 pmol/mol, which is consistent with winter mean concentrations above boreal forests reported for the years 2000-2007 (Hakola et al., 2009). On the other hand, for the year 2016 wintertime β-pinene concentrations were reported lower (~1 pmol/mol) at the same site (Hellén et al., 2018). In conclusion, we have to state that the mean and median concentrations for the short lived biogenic compounds in Figure 6 m-p are at the edge of the detectable range for our instrument. Therefore we cannot robustly claim the detection long-range transport of monoterpenes and their oxidation products. However, we note that Fu et al. (2009) quantified several oxidation products from monoterpenes and sesquiterpenes in weekly high-volume aerosol samples from Alert (Canada, 82.5° N, 62.3°





W) in the period February to June, 1991. Especially during winter and spring, they quantified several oxidation products

from monoterpenes and sesquiterpenes, while the samples from June were dominated by isoprene oxidation products.

The data presented in table 1 and in Figure 6 show that relatively stable biogenic emissions are transported to the high

Arctic, and that the same may be true for isoprene, monoterpenes, and their oxidation products. Emission and photochemical

degradation of terpenes (i.e. monoterpenes and sesquiterpenes) are very important for the production of organic aerosol. In

order to further explore whether terpene oxidation products are present in the high Arctic atmosphere we exploit a study

from Lee et al. (2006). These authors inventoried 73 ions measured by PTR-MS that can be associated with the

photooxidation of 11 monoterpenes, 4 sesquiterpenes, and isoprene. Figure 7 shows the fraction of measurements (n=7771)

above the LOD for all detected ions above 40 Th (n=318). 66% of the ions associated with terpene oxidation products were

above the LOD for more than 3% of the measurements. In contrast, only 23% of all other ions were detected above the LOD

for more than 3% of the measurements. Of the 104 ions that complied with the latter criterion almost half (48 or 46%) can be

associated with terpene oxidation products. This comparison may be an indication of a strong signature of biogenic terpenes

and their oxidation products in the high Arctic atmosphere, however, more research is certainly needed to quantify and

assess its impact.

Table 2 reproduces the information given in Figure 7 for the 73 ions identified by Lee et al. (2006) and gives the mean

concentration that we measured during the entire campaign (~25 flight hours). To assess the SOA formation potential of

these compounds in the high Arctic we consider only compounds heavier than protonated monoterpenes, because these

heavier oxidation products efficiently condense on particles. Summing up all compounds heavier than 138.5 Th we conclude

that, in average, the Arctic atmosphere contained ~12 pmol/mol of biogenic oxidation products that are expected to partition

efficiently into the condensed phase. Considering the limitations of the inlet system and the zeroing method, we think that

this is rather a lower limit and that the real content of condensable organics could be larger. Note that the sum mixing ratio of

these presumed oxidation products is similar to the measured levels of DMS (see Table 1), and that the mass density of these

compounds is typically 20% of the mass of non-BC particles measured during the campaign (size range 185-850 nm, typical

volume 0.4 $\mu m^3$ $cm^{-3}$). Both these figures show that transported terpene oxidation products have the potential to be a key

player in controlling the interaction between clouds and radiation in the high Arctic winter. The impact of terrestrial

vegetation on the Arctic atmosphere could be substantial because the photochemical degradation of plant volatiles provides

material that can be converted to organic aerosol. In this way, the vegetation exerts influence on the complex system of

aerosol, clouds and radiation in the Arctic climate system. There are two aspects that highlight the importance of this finding.

Firstly, biogenic SOA could have been a major factor controlling aerosol/cloud/radiation interaction in the Arctic winter

before aerosols from pollution took this role. Secondly, the ongoing climate change will change boreal ecosystems and their

emissions, while pollution will – hopefully – be cleaned up in the near future. So, biogenic SOA likely re-gain importance in

the future.



## 4 Summary and Conclusions

We analysed a dataset containing 7771 mass spectra (containing 353 ions), each averaged over 10 seconds, that were obtained with a PTR-MS during 10 flights with the POLAR 5 research aircraft in the high Arctic covering a region between

50-4000 m above sea level, and 78°N -85°N, 20°W -20°E. The analysis was aided by back trajectory calculations, and measurements of CO, $CO_2$, $H_2O$, ozone, and particles. We were able to show that continentally influenced air contained significantly enhanced levels of aged biogenic and anthropogenic emissions (e.g. methanol, PAN, acetone acetic acid, MEK, proprionic acid, and pentanone) when compared to periods that were classified as Arctic/marine.

During two flights, we encountered ODEs above the sea ice. These events were associated with enhanced acetone, MEK and

pentanone mixing ratios, in accordance with findings by Yokouchi et al. (1994), Guimbaud et al. (2002), and Hornbrook et al. (2016). During the same two flights, Hartmann et al. (2020) observed largely enhanced INP concentrations with onset temperatures as high as 265.65 K, which points towards biological INP, i.e. macromolecules that promote freezing of condensing water at high temperatures. The co-occurrence with the ODEs suggests that these biological INP might be emitted along with bromine that causes the ODEs.

An analysis of the full mass spectrum revealed that almost half of the ions that were above the LOD in at least 3% of the measurements can be associated with oxidation products of terpenes. The presence of terpene oxidation products constitutes a link to organic aerosols because these are well known aerosol precursors. Altogether, our results suggest that long-range transport of biogenic emissions in winter and spring provides a natural source of cloud forming aerosol particles in the high Arctic independent of the pollution that is transported to this region as well. The sum concentration of the detected aerosol

forming vapours is ~12 pmol/mol, which is of the same order than measured DMS mixing ratios and the mass density corresponds to approximately one fifth of the measured non-BC particles. This natural aerosol source existed also before the industrial revolution and could be an important control of the interaction between aerosols, clouds, and longwave radiation during the high Arctic winter.

**Acknowledgements**

Special thanks go to Gernot Hanel and co-workers from Ionicon Analytic GmbH, who assisted during critical trouble shooting after office hours, which was successful despite the poor quality of uncounted satellite voice-only calls connecting North Greenland with Innsbruck, Austria. We gratefully acknowledge the funding by the Deutsche Forschungsgemeinschaft (DFG, German Research Foundation) – project ID 268020496 – TRR 172, within the Transregional Collaborative Research

Center "ArctiC Amplification: Climate Relevant Atmospheric and SurfaCe Processes, and Feedback Mechanisms (AC)3".



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






Table 1: Mean mixing ratios of compounds during different periods classified as 'continental', 'Arctic/marine', and 'sea ice/ODE'. **Bold, cursive numbers** are significantly different from the other two groups (95% confidence).

| Detected M/q | Possible attributions | Mean mixing ratio ± standard deviation in pmol/mol | | |
| --- | --- | --- | --- | --- |
| | | Continenta | Arctic/marine | Sea ice/ODE |
| 33.021 | methanol | *426 ± 92* | 152 ± 33 | 153 ± 38 |
| 43.011 | $C_2H_3O^+$ | *95 ± 36* | 34 ± 5.4 | 36 ± 5.1 |
| 45.987 | $NO_2^+$, mostly PAN | *187 ± 78* | 70 ± 6.9 | 61 ± 10 |
| 47.013 | formic acid | 64 ± 38 | 35 ± 17 | 23 ± 4.8 |
| 57.065 | $C_4H_9^+$ | 36 ± 7.4 | 32 ± 5.7 | 33 ± 5.0 |
| 59.050 | acetone | 887 ± 124 | *453 ± 88* | 733 ± 69 |
| 61.030 | acetic acid | *57 ± 24* | 16 ± 3.4 | 18 ± 5.9 |
| 63.009 | DMS | 13 ± 2.9 | 14 ± 7.0 | 9 ± 1.5 |
| 69.069 | isoprene | *17 ± 2.8* | 12 ± 2.6 | 10 ± 2.0 |
| 73.062 | MEK | 174 ± 39 | *96 ± 18* | 172 ± 15 |
| 75.043 | propionic acid | *17 ± 6.3* | 9 ± 2.0 | 9 ± 0.7 |
| 79.055 | benzene | 66 ± 19 | 61 ± 2.7 | 60 ± 3.1 |
| 83.081 | $C_6H_{11}^+$, cis-3-hexen-1-ol | *12 ± 2.0* | 6 ± 2.7 | 4 ± 1.3 |
| 87.055 | pentanone | 24 ± 7.7 | *15 ± 2.0* | 23 ± 3.0 |
| 93.070 | toluene | 11 ± 4.3 | 9 ± 2.0 | 7 ± 1.1 |






Table 2: Mean mixing ratio and fraction above the limit of detection for all 73 mass to charge ratios (m/Q) at which Lee et al. (2006) detected terpene oxidation products. Values below 3 % are plotted in grey.

| m/Q [Th] | mean, pmol/mol | above LOD | m/Q [Th] | mean, pmol/mol | above LOD | m/Q [Th] | mean, pmol/mol | above LOD |
|---|---|---|---|---|---|---|---|---|
| 57.023 | 4.9 | 7% | 100.041 | 0.3 | 2% | 141.085 | 0.6 | 4% |
| 59.05 | 561 | 95% | 101.033 | 2.7 | 8% | 142.028 | 0.0 | 0% |
| 60.049 | 18 | 81% | 102.036 | 0.0 | 1% | 143.109 | 1.2 | 6% |
| 63.009 | 9.6 | 21% | 103.039 | 0.2 | 2% | 145.066 | 0.3 | 1% |
| 65.019 | 3.2 | 19% | 105.049 | 1.0 | 4% | 149.054 | 1.8 | 5% |
| 69.069 | 15 | 63% | 107.047 | 2.4 | 3% | 151.131 | 0.6 | 4% |
| 71.053 | 4.5 | 13% | 108.072 | 1.9 | 4% | 153.098 | 0.4 | 3% |
| 72.046 | 0.6 | 3% | 109.089 | 4.8 | 20% | 155.115 | 0.3 | 2% |
| 73.062 | 118 | 95% | 110.094 | 0.4 | 2% | 157.125 | 0.4 | 1% |
| 74.033 | 4.3 | 24% | 111.113 | 5.6 | 34% | 159.093 | 1.0 | 5% |
| 75.043 | 9.2 | 49% | 113.041 | 0.9 | 5% | 163.141 | 1.9 | 3% |
| 77.023 | 1.2 | 8% | 113.112 | 1.1 | 6% | 165.085 | 0.2 | 2% |
| 79.055 | 63 | 95% | 115.05 | 2.6 | 12% | 165.157 | 0.6 | 4% |
| 81.062 | 3.8 | 22% | 121.065 | 4.7 | 9% | 169.061 | 0.2 | 2% |
| 82.057 | 0.2 | 1% | 122.091 | 1.4 | 10% | 171.091 | -0.2 | 0% |
| 83.081 | 7.9 | 38% | 123.097 | 1.4 | 8% | 178.093 | 0.2 | 2% |
| 85.056 | 1.5 | 3% | 125.119 | 1.7 | 9% | 181.079 | 0.5 | 4% |
| 86.059 | 0.1 | 2% | 129.087 | 0.7 | 3% | 183.089 | 0.3 | 2% |
| 87.055 | 16 | 72% | 131.05 | 0.3 | 0% | 187.084 | 0.3 | 3% |
| 93.07 | 10 | 56% | 133.022 | 0.0 | 1% | 201.097 | 0.0 | 0% |
| 95.044 | 2.2 | 6% | 133.067 | 0.8 | 2% | 205.101 | 0.2 | 1% |
| 97.094 | 3.8 | 16% | 135.084 | 3.2 | 7% | 205.184 | 0.4 | 4% |





| | | | | | | | | |
|---|---|---|---|---|---|---|---|---|
| *98.1* | *0.2* | *2%* | *137.123* | 1.4 | 7% | *229.191* | *0.1* | *1%* |
| *99.019* | 3.6 | 18% | *138.119* | 0.2 | 3% | | | |
| *99.076* | 2.1 | 10% | *139.129* | 0.6 | 3% | | | |


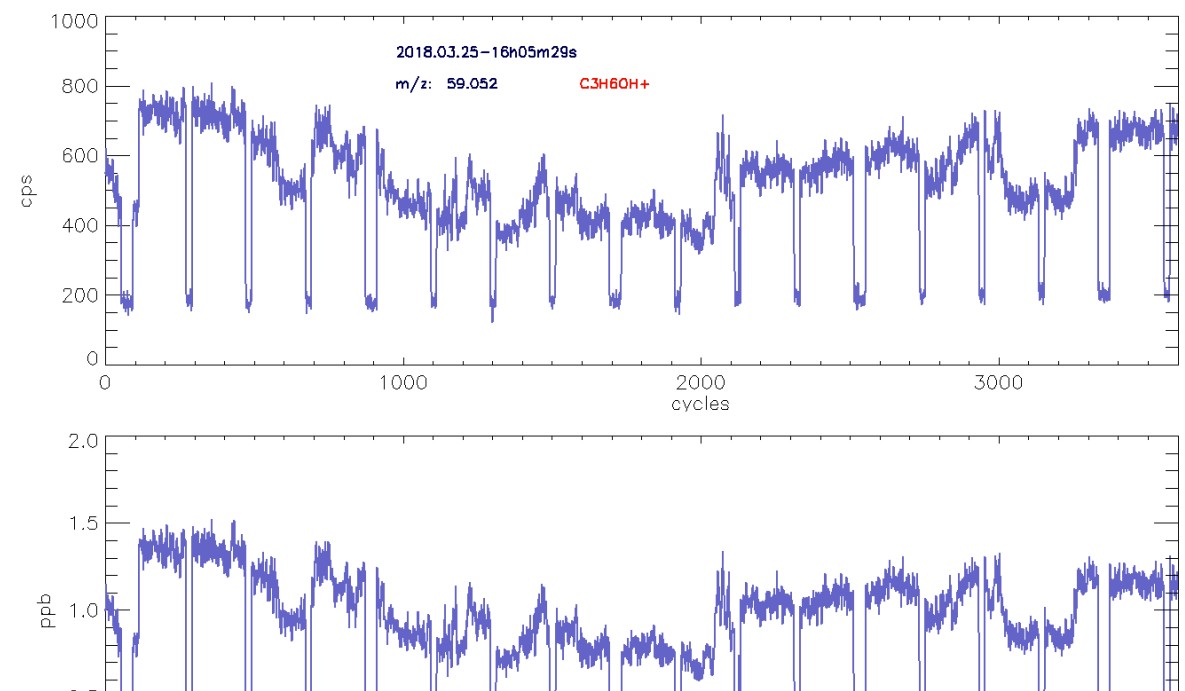


**Figure 1:** Output of the PTRwid software (Holzinger 2015) for the raw signal at 59.052 Th (acetone) during one hour of flight 2 (March 25, 2018). Three minutes of ambient air sampling are intercepted by periods of zeroing (3 times 20 s, and 40 s every 4th time). Volume mixing ratios are calculated based on simple ion reaction kinetics (Holzinger et al., 2019) using a default reaction rate constant of 3 x 10$^{-9}$ cm$^3$ s$^{-1}$ molecule$^{-1}$. The data were recorded at a 1 Hz frequency and the instrument featured a sensitivity around 500 counts per second 505 (cps) for a compound at 1 nmol/mol (ppb).



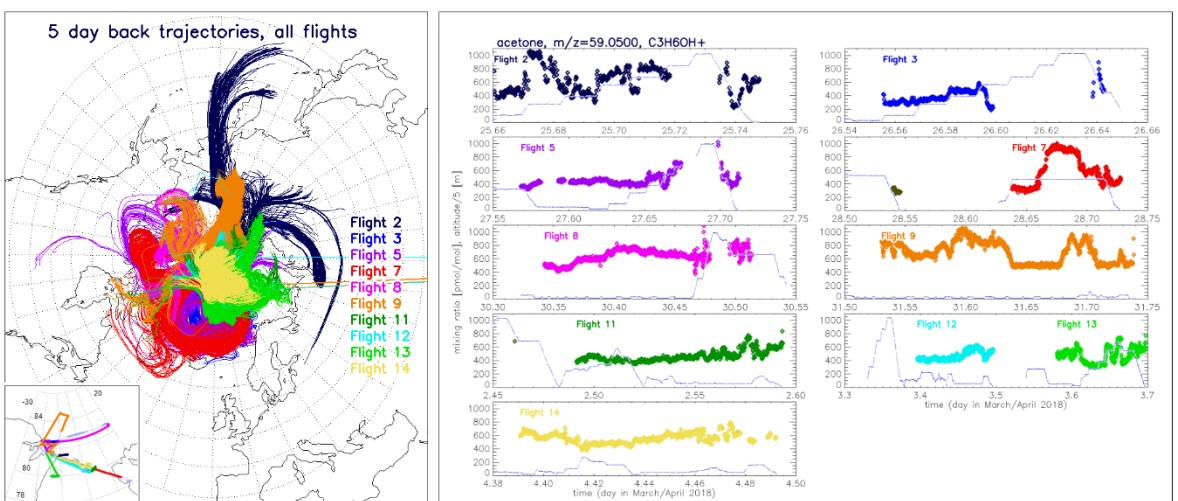

**Figure 2:** The left panel shows 5 day back trajectories during all periods that are selected for the analysis color-coded by flight number. Trajectories were calculated every 10 seconds. The bottom-left insert shows the flight paths. The right hand panels show acetone mixing ratios and altitude above sea level for all flights.

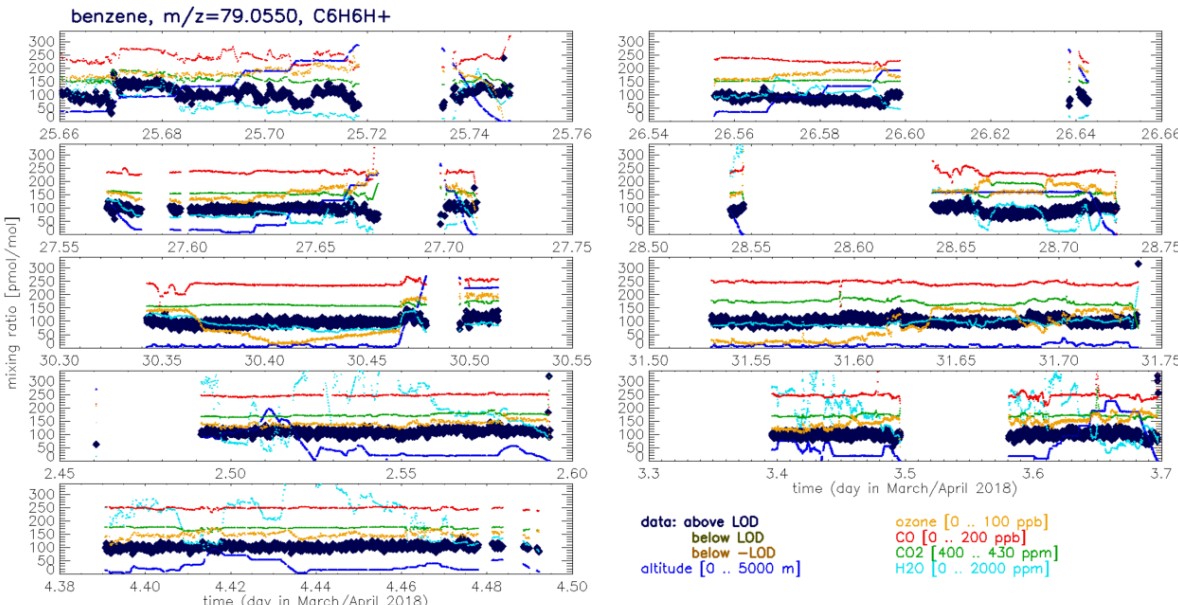

**Figure 3:** Measured mixing ratios for benzene for all flights together with mixing ratios of ozone, CO, $CO_2$, and $H_2O$, as well as the altitude above sea level. A y-axis is only drawn for benzene mixing ratios. The scaling factors plotted in brackets for the other datasets correspond to the range that is covered by the charts. All benzene data are plotted in black, which means that all measurements were above the limit of detection (Figure S1 shows an example where this is not the case).






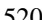

**Figure 4:** Case studies during flight 2 (top panels) and flight 7 (bottom panels). The left/right hand panels show back trajectories and mixing ratios of acetone and other trace gases, respectively. The top left panel shows 7 day back trajectories for the periods marked by green, blue, and magenta bars on the x-axis of the top right panel. The trajectories are plotted in a 'fading' color scale that corresponds to the altitude above ground (note the different altitude scales in the color-legends). The trajectories corresponding to the green bar in the top right panel start at less than 1000 m above the surface in northern Finland. Flight 7 was a transfer flight from Longyearbyen, Svalbard, to Station Nord, Greenland. The trajectories marked by the green bar in the bottom right panel correspond to the cluster that moves at low altitudes across the North Atlantic and Greenland. See main text for further interpretation.






**Figure 5:** Case studies concerning flight 8 (top panels) and flight 9 (bottom panels) during which ozone depletion events have been encountered. Same color coding as described in the caption of Figure 4.





**Figure 6:** Average (connected circles) and median (horizontal bars) concentrations of anthropogenic and biogenic components categorized in bins of increasing acetone mixing ratios: 300-400 (n=1109), 400-500 (n=2284), 500-600 (n=1503), 600-700 (n=613), 700-1000 (n=590) pmol/mol, respectively. Data exhibiting ozone levels below 30 nmol/mol have been removed from this analysis to avoid biases from ODE events.




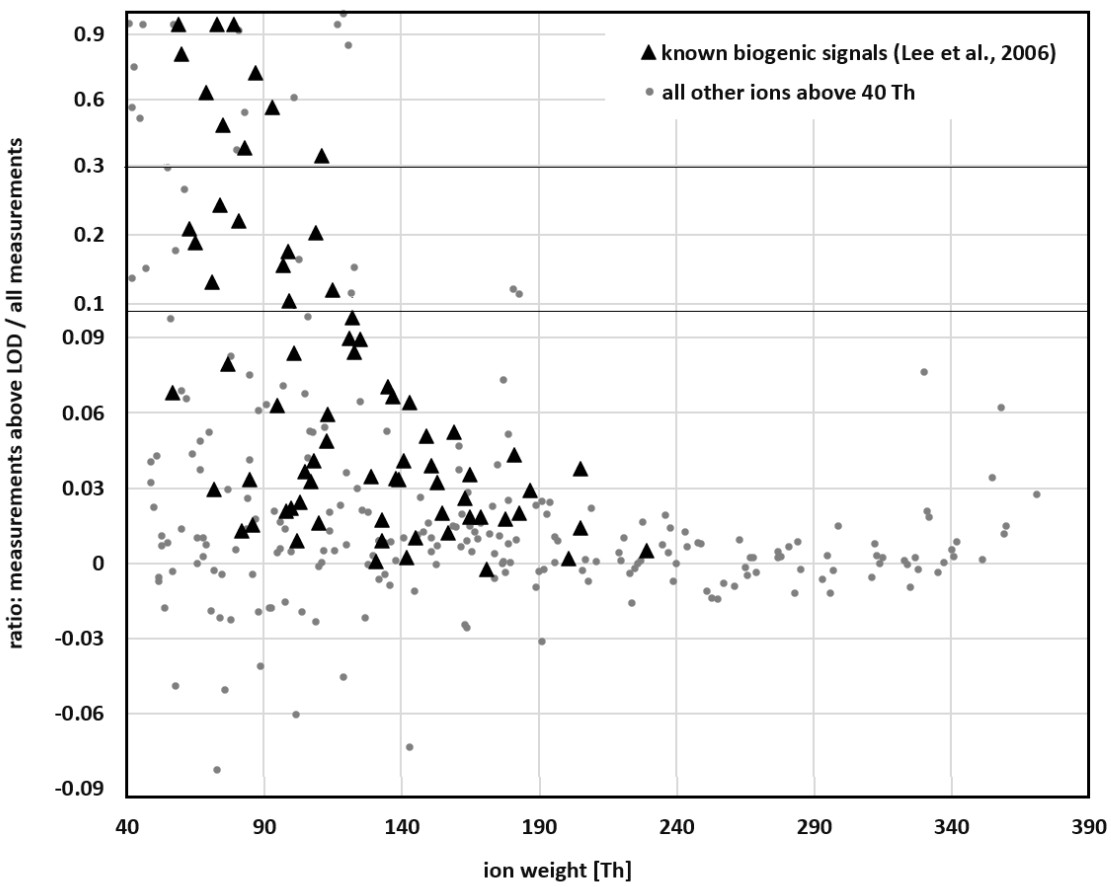

**Figure 7:** Fraction of measurements above the limit of detection for all 318 compounds above 40 Th. Terpene oxidation products from Lee et al. (2006) are plotted in black triangles. Negative values indicate compounds that exhibited higher concentrations during the zeroing cycle (contamination). Compounds without clear signal in ambient air are scattered around the zero line.