# Peer review of "Possible controls on Arctic clouds by natural aerosols from long-range transport of biogenic emissions and ozone depletion events"

_Atmospheric Chemistry and Physics, 2022_

## Author Comment (AC1)

**Response to reviewer comments on n "Possible controls on Arctic clouds by natural aerosols from longrange transport of biogenic emissions and ozone depletion events" by Rupert Holzinger et al., Atmos. Chem. Phys. Discuss., https://doi.org/10.5194/acp-2022-95-RC2, 2022**

We thank two anonymous reviewers for their useful comments.

First, we need to acknowledge that the title, parts of the abstract, and the conclusions raised expectations that cannot be satisfied by the work presented here. We acknowledge that this was a miscommunication on our end and we hope to fix this by changing the title to "", and by adjusting the relevant parts in the abstract and the conclusions. For clarity, we point out the key findings of this study here before we continue with a point-to-point reply of specific reviewer comments below.

Key findings:

➔ Air with enhanced levels of continental influence carried the fingerprint of oxidized biogenic emissions.

➔ High concentrations of ice-nucleating particles (INP) are possibly linked with ozone depletion events (ODEs).

These are important findings that, in our opinion, make a valuable contribution to ACP. The possible implications that were highlighted (too much) in title/abstract/conclusions of the previous version warrant follow-up studies as both the reviewers suggest.

**Point-to-point reply (reviewer comments are duplicated in *"cursive"* letters):**

**Reviewer 1:**

**Major comments**

**Comment 1:** *"The title is totally misleading. … I suggest to reformulate the title focusing on what is actually done here and of sufficient interest : evidence of the signature of biogenic compounds and their oxidation products in the Arctic atmosphere. The authors may also want to mention ODEs that have indeed been detected in their observations."*

We changed the title to *'Lessons from airborne VOC measurements during PAMARCMIP 2018 concerning aged biogenic emissions and ozone depletion events'*.

**Comment 2:** *"Abstract : L 38-47 : This section is rather speculative and should be removed. It is a rather a sequence of hypotheses (influence on clouds, link between ODEs and INPs) that are not supported by the analyses performed here. They shouldn't appear as key results in the abstract and conclusions. In addition, the effect of ice crystals (that could be linked to INPs, but also to secondary ice production) is much lower than that of supercooled droplets in the longwave spectrum."*

We removed interpretations and stick to what was actually observed.

**Comment 3:** *"L 43 : "evidence that ODEs may be". Is it a conclusion or an assumption ?"*

Neither conclusion nor assumption. The observations clearly show that we crossed ODE events during these flights. During each flight only one filter was sampled for INP analysis. The filters were sampled during the entire flight duration (except start and landing to avoid contamination) and exhibited enhanced INPs for the two flights that crossed ODEs. Theoretically it is possible that the INPs were collected outside the regions with ODE, but the co-occurrence of ODEs and enhanced INPs is a strong evidence that these two are linked.

**Comment 4:** *"L 318-326 : This section is written in a qualitative style without any analysis to support this discussion. In addition, biogenic SOA are presented as a "major factor controlling aerosol/cloud/radiation interaction in the Arctic winter" without mentioning which physical process is concerned. Do the authors want to specify their role as CCN versus their role as INP ? Do they consider their role on SW and LW radiation ? Which minimal optical properties are required to distinguish the effects ?"*

We improved the discussion by (i) focusing on the primary process (the production of secondary organic aerosols), (ii) not highlighting secondary effects (on clouds and radiation), and (iii) acknowledging that more research is needed before such claims can be solidified.

**Comment 5:** *"Summary/conclusions : Some parts of the conclusions are not justified in the study. A large fraction of the summary (L 334-339) is taken from results from other studies, not from this study itself. None of biological INP or bromine species are measured/analysed here, but the authors nevertheless give conclusions relative to their co-occurrence with ODEs."*

We partially disagree with the reviewer because we reference Yokouchi et al. (1994), Guimbaud et al. (2002), and Hornbrook et al. (2016) just to state that our results are in agreement with their findings. The INPs have indeed been reported by Hartmann et al. (2020), however, they did not make the link with the ODEs. We think that the possible link between enhanced INPs and the ODEs is important and must be reported. In hindsight, it would have been better if this association had been realised (and reported) already in the Harmann et al. paper. So, in this aspect we agree with the reviewer, but we do not see a better solution.

**Comment 6:** *"- labels are difficult to read on Fig. 2. Use larger/bold font to make this clearer."*

Labels are in bold font now.

**Comment 7:** *"- Fig 3, 4, 5 : It is extremely difficult (or impossible) to read as the scales for the different components are not plotted. The reader has to convert himself/herself the units to understand the variability of the different species. This is not acceptable in a published paper. Please add values on the y axis or add different sub-figures if there are too many curves. In any case, the reader should be able to read easily the values of all the different species that are plotted."*

We removed CO, $CO_2$, and $H_2O$ and kept only the essential datasets: VOCs, flight altitude, and ozone. This should make the Figures more accessible. CO, $CO_2$, and $H_2O$ are still shown in the supplemental Figures.

**Comment 8:** *"- Fig 6 : This figure only give mean and median values for the different components. Boxplots including the different quartiles would provide more useful information about the variability of the mixing ratios (not concentrations, as written) of the various components for each bin of the acetone mixing ratios."*

It was a considerate choice to show only median and average in Figure 6, so that the enhancements are immediately visible. However, we agree with the reviewer that information about variability would give a more complete picture. Therefore we provide the same Figure with standard deviations as supplemental Figure S15.

**Specific comments:**

**Comment 9:** *"L 50-58 : This is a strange way to start a manuscript with no reference at all. A classical introduction starts by a state-of-the-art of the scientific literature. Please add some useful references that support the discussion."*

References added.

**Comment 10:** *"L 58 : "anthropogenic pollution became the dominant control". Is there some evidence for that ? All INPs that are active in Arctic mixed-phase clouds are known to be of natural origin (mostly dust or biologic macromolecules) : see for examples the recent papers of Wex et al. (2019, https://doi.org/10.5194/acp-19-5293-2019), Kanji et al. (2020, https://doi.org/10.1029/2019GL086764), Welti et al. (2020, https://doi.org/10.5194/acp-20-15191-2020). Do the authors rather want to underline the influence of anthropogenic aerosols on liquid clouds here ? This is not clear."*

Yes. This statement is about aerosol pollution and their role as CCN. This has been clarified in the text now.

**Comment 11:** *"L 62-76 : The two studies quoted here (Nielsen et al. , 2019 and Pernov et al., 2021) report measured at the surface. What are the connections those observations and layers containing aerosol and trace gases aloft that are sampled by an aircraft ? Are there vertical profiles available ? "*

These papers report measurements at the surface but both cover a period of several months. We assume that measurements at a site without dominant local pollution over such long periods capture the regional features of the atmosphere. While Nielsen et al. (2019) measured the composition of aerosols and Pernov et al. (2021) measured VOCs, they both extract very similar chemical regimes with the Arctic haze factor dominating until ~end of April, and local/marine processes taking over in spring and summer. No vertical profiles were available.

**Comment 12:** *"L 79 : What are the scientific questions of this study ? What is the outline of the paper ?"*

Following paragraph has been added: 'The main goal of this work is identifying possible long range transport of biogenic emissions to the high Arctic troposphere by exploiting full mass spectra obtained with a PTR-TOF-MS instrument.'

**Comment 13:** *"L 152 : Which ECMWF data are precisely used in the study ? Are they reanalyses (ERAInterim or ERA-5) ?"*

The LAGRANTO model uses ERA-interim.

**Comment 14:** *"L 153 : How many trajectories are initialized every 10 s ? Is there only 1 trajectory initialized every 10 s along the flight track ? A series of boxes sliding along the flight track with multiple trajectories in each of them would give an idea of the uncertainties and of the dispersion."*

Only one trajectory has been initialized every 10 seconds. We acknowledge that the trajectory model could be exploited more profoundly to corroborate our air origin classification and this would make a lot of sense if our classification was based on the trajectory analysis alone. However, we use a chemical tracer (acetone) as first parameter. The trajectories are used in a few case studies to show that the general interpretation of the chemical tracer is consistent with the air flow regimes.

**Comment 15:** *"L 157-158 : Some of the trajectories are located over the Eastern Coast of the US with potential large anthropogenic emissions; others are over Siberia or Canada where intense fires are detected every year ; some may also originate from anthropogenic soures in East/South Asia if they were ploted for periods larger than 5 days. It is therefore difficult to conclude something from this Fig. 2 as we don't have any information about the altitude of the backtrajectories at this stage, nor on th potential deposition during transport."*

We point out that the trajectories are not our main tool to classify air mass origin/contamination: The absence of biomass burning plumes is clear from the low levels of acetonitrile. Benzene is a tracer for anthropogenic emissions and biomass burning, while acetone is a tracer of all aged emissions, biogenic, anthropogenic and biomass burning.

The left panel of Fig 2 provides an overview of 5 day back trajectories along all flight tracks. It is not feasible to show additional information (altitude, deposition) without making the Figure unreadable. Altitude information is provided for the trajectories shown in Figures 4 and 5. For extra information, we show 10 days back trajectories below.

[Figure]

**Comment 16:** *L 175-176 : How do the authors disentangle the influence of the origins of air masses from that of wet/dry deposition mechanisms along transport ?*

Acetone is not considerably removed by wet/dry deposition. So, these processes are irrelevant for acetone and most other VOCs. However, it is possible that, for example, aerosols are removed by precipitation events during the transport. However, the good correlation between and acetone (Figure 6b, and 6c) indicates that deposition was not dominant enough to wipe out the shared source signature.

**Comment 17:** *L 190 : Do the authors mean that air masses can be influenced by pollution only when trajectories are observed in the vicinity of the surface ? What about the mixing at higher altitudes with polluted air masses (anthropogenic or biomass burning) ?*

We are well aware of the fact that biomass burning emissions can be injected at high altitude due to local deep convection. However, biomass burning impact can be excluded by the low levels of acetonitrile. In L190 we discuss a short feature of low acetone levels and trajectories that show now

evidence of surface contact. The periods before and after this episode exhibit higher acetone levels and these trajectories indicate surface contact.

**Comment 18:** *L 219-220 : Yes, halogen chemistry is an active field of research ; but the references given here are rather "old" to justify this. More recent references would be helpful here.*

We narrowed the statement and now refer to the production of oxygenated compounds. There are only few studies. Hornbrook et al. (2016) is actually the newest that we could find….

**Comment 19:** *L 232-242 : This discussion is interesting but outcomes are from other studies and the co-emission of biological INPs is only an hypothesis here. I think it is worth mentioning this here, but not in the conclusions of the paper. Otherwise, I suggest an investigation of this effect, with at least correlations.*

We agree that the co-emission of biological INPs during ozone depletion events is a hypothesis at this stage. But our measurements (that were partially reported in Hartman et al., 2021) show evidence for this hypothesis: two of the three flights during which enhanced INP were observed during the two flights with the ODEs. The proof of this hypothesis needs to come from future measurements. Note that during each flight only one INP filter sample was taken. Therefore it is hard to show any correlations.

Also see our replies to comments 3 and 5.

**Comment 20:** *L 318 : "Both of this figures show" : I disagree, I don't see any evidence for that on those figures.*

We changed the sentence to "Both these numbers show…" to avoid misunderstandings. The fact that the terpene oxidation products are of similar order as measured DMS and aerosol concentrations, clearly demonstrates their potential importance.

**Comment 21:** *L 331-333 : May the role of precipitation during transport influence of result ? I don't think it has been considered here, but precipitation amounts along trajectories (drizzle vs large scale precipitation) are available in the ECMWF analyses used by the authors.*

Precipitation certainly exhibits influence that is worthwhile studying. However, the volatility of the compounds mentioned in L331-333 is relatively high. So, they are expected to remain predominantly in the gas phase also during precipitation events.

Also see our replies to comment 16.

**Comment 22:** *L 343 : " a natural source of cloud forming aerosol". I don't see any evidence of this in the paper. Even if biogenic emissions may produce aerosols that can serve as INPs, there is no evidence that they actually do it and that ice crystals are formed at this period in this region.*

The term "cloud forming aerosol" does not suggest that these particles are INPs. But if their size exceeds ~90-100 nm, they will be effective cloud condensation nuclei (CCN).

Secondary organic aerosol formation from aged biogenic emissions has been studies in many laboratory and field settings, and the result is always a production of CCN. So, if we believe that we detected aged biogenic emissions in the high Arctic, we can conclude that these emissions are also a natural source of CCN, i.e. cloud forming aerosol.

**Technical comments:**

All technical comments have been fixed.

**Reviewer 2:**

**Comment 1:** *"The hypotheses presented in the paper are not supported by the data and plots included. For example, there is no demonstrated link between clouds, the biogenic emissions, natural aerosols, and ozone depletion events as suggested by the title. This is acknowledged in the title with the word "possible". The information presented in the paper does not suggest even possible controls, as noted by reviewer 1."*

We changed the title and we refer to our introductory statement of our replies. We acknowledge that the title and parts in the abstract and conclusions triggered expectations that are not within the scope of this work.

**Comment 2:** *"There is no data availability section. This means that the only information that can be concluded from the paper is from the very poor presentation of the data in the plots and tables provided. From the figures and tables alone, I find it nearly impossible to follow the arguments presented. For example, the trajectory analysis is impossible to read on the plots. Most of the plots are time series of the PTR-MS data of specific chemical compounds for each flight. The data should be provided in a public repository or as a publication in ESSD and the authors should complete a more thorough scientific analysis prior to re-submission to ACP."*

We made the data available at "https://surfdrive.surf.nl/files/index.php/s/M9iq1O2T0ftNn9o". After final acceptance the data will be transferred to the UU repository YODA and publicly available with a unique digital object identifier (doi).

**Comment 3:** *"Neither the "Introduction" or the "Summary and Conclusions" section provides an accurate review of what has been done in past work or an analysis of what scientific conclusions are new that are presented in this study. What I see here is that the authors try to mainly show PTR-MS data along with air trajectories. This is an incomplete scientific story and is more adapted for a data publication journal (such as ESSD)."*

We acknowledge the miscommunication as stated above, but we respectfully disagree with the general notion of this statement. Our analysis clearly identified the fingerprint of biogenic oxidation

products in air that was impacted by continental emissions. Moreover, we analysed ozone depletion events and we could confirm the production of several VOCs during these events and there is also evidence that ODEs could be a source of INPs.

**Comment 4:** *"Each of the sub-sections in the "Results and Discussion" section could be its own scientific paper with an in-depth analysis of the data presented here along with all of the other relevant to atmospheric chemistry collected on the POLAR5 to show: (1) anthropogenic pollution influence on Arctic Haze during March and April 2018; (2) air mass characteristics in pristine air originating from the Arctic surface (sea ice/ocean sources); (3) characterics of air masses that have biogenic influences (separating marine and terrestrial sources). Each of these papers would require the authors to go much further than what is presented here to make scientific conclusions from the data."*

We appreciate these suggestions, but we think that they go far beyond the scope of this work. These are rather suggestions for future research. The 2018 PAMARCMIP campaign included a limited set of instrumentation in one research aircraft and ground based measurements at one location (Station Nord). The presented results are, in our opinion, a valuable scientific contribution.